# Bi-directionality in translating culture: Understanding translator trainees' actual and perceived behaviors

**Mutahar Qassem[1]\*, Buthainah M. Al Thowaini[2]**

**1** Department of Translation, Najran University, Najran, Kingdom of Saudi Arabia, **2** Department of English Language, King Saud University, Riyadh, Kingdom of Saudi Arabia

\* mutaharnahari@gmail.com

**Data Availability Statement:** All relevant data are within the manuscript and its Supporting Information files. The data include xml files that are extracted from Translog software for each participant. I have shared minimal data in the

## Abstract

Cultural translations of L1 and L2, in both directions, can indicate different behaviors of translators, influenced by the unique characteristics of each culture and the proficiency of the translator trainees' bicultural competence. This study compares translator trainees' behaviors when engaged in direct translation (L2 to L1) and inverse translation (L1 to L2) of cultural references to reveal the extent to which directionality influences trainees' actual and perceived behaviors. Following a hypothesis-based observational design, the authors examine a single group's behaviors under two conditions (direct translation and inverse translation), using Translog-II and a questionnaire. The data are analyzed quantitatively using the Wilcoxon test and descriptive statistics. The key findings indicate that inverse translation demands more cognitive effort than direct translation, particularly in online revision ($n = 16$, $z = -3.206$, $p < .05$) and production speed ($n = 16$, $z = -3.068$, $p < .05$). Conversely, direct translation requires more cognitive effort, especially in orientation time ($n = 16$, $z = -2.482$, $p < .05$) and performance ($n = 16$, $z = -3.346$, $p < .05$). Additionally, the students' responses to the questionnaire reveal a tendency to rely more on online resources than on internal translation strategies. The authors suggest that translation students should receive training in both translation directions, effective management of the translation process, appropriate utilization of translation strategies, and cultural competence. These components should be integrated into translation training courses and instructional methods.

## Introduction

The translation of culture poses challenges for professional and novice translators alike. These challenges arise from the different cultural landscapes of the source language (SL) and the target language (TL), as well as the bicultural competence of translator trainees. Scholars have explored the issue of culture from different perspectives, such as cultural studies, discourse analysis, contrastive analysis, and translation studies [1–4] Given the extent and complexity of culture, this study focuses specifically on a singular aspect of translation–cultural references. These references encompass words and phrases in the source culture that do not have equivalents in the target culture, thereby having different denotations or connotations in the two

Supporting Information file entitled "Strategies survey- (Responses).xlsx" and "Strategies survey-(Responses).rar".

**Funding:** The Authors would like to acknowledge the support of the Deputy for Research and Innovation- Ministry of Education, Kingdom of Saudi Arabia for this research through a grant (NU/IFC/2/SEHRC/-/12) under the Institutional Funding Committee at Najran University, Kingdom of Saudi Arabia.

**Competing interests:** The authors report no conflicts of interests.

contexts. Essentially, they embody expressions signifying any tangible, environmental, societal, spiritual, political, linguistic, or emotional manifestation tied to a particular community [5].

In Saudi Arabia's academic environment, there is a recognized need for proficient translators. To address this demand, universities have introduced specialized translation training programs at both graduate and postgraduate levels [6]. These programs aim to enhance the translation skills of trainees, encompassing bi-cultural competence as a crucial component of their overall translation competency. Our research explores these trainees' behaviors during the translation of culture-based texts from English to Arabic and Arabic to English, utilizing Translog software to record these behaviors. By closely examining their behaviors during the translation process, we strive to identify areas that require improvement. Our ultimate goal is to propose effective solutions that can improve translators' ability to handle cultural references more adeptly.

## Theoretical framework

The study's theoretical framework comprises four topics: actual translation behaviors, perceived translation behaviors, translation directionality and the translation of cultural references. Each of these is discussed below.

## Actual translation behaviors

Researchers study translation behaviors to diagnose the strengths and weaknesses of those behaviors, using questionnaires, translation tests, and think-aloud protocols to reveal the internal and external strategies applied during the translation process [7]. With the advancement of technology, researchers began using software, such as keystroke logging data software and eye-tracking, that is capable of tracking the subjects' translation behaviors [8].

Various models have been developed to explain the cognitive processes involved in translation production, drawing on Cognitive Psychology and Information Processing frameworks [9–14]. Of such models, one of the most influential classifies the translation process into source-text (ST) input, target-text (TT) output, and interaction between ST input and TT output [15]. Jakobsen and Schou [16] introduced a three-stage empirical model for the translation process comprising: initial orientation, drafting, and final revision. To document these stages, they employed Translog software that can capture cognitive behaviors during the three stages of translation and shows the participants' management of these stages in terms of time allocation, revision activities, and utilization of online resources [17]. The software (using pause metrics) records the amount of time allocated to orientation, drafting, and revision; tracks revision activities through deletion keystrokes; and monitors the use of online resources via navigation keystrokes. These parameters serve as indicators of the cognitive efforts expended during the translation process [18].

## Bi-directionality in translation

This study addresses the translation of cultural references in terms of directionality, which pertains to the translation of texts from a second language(L2) into one's first language (L1) or vice versa (L1→L2) [19]. Directionality has been explored in several studies from various perspectives such as cognitive processes, utilization of online resources, metaphor translation, and translators' viewpoints [20–24]. Proficiency in directionality is a skill that is essential for translators, with several researchers emphasizing its importance [25–28]. This current study makes a novel contribution by exploring the directional translation of cultural references using keylogging data.

The issue of directionality has been investigated in several studies, albeit from different angles. The general assumption is that inverse translation is more difficult, time-consuming and effortful, although many empirical studies report results that are inconsistent [20, 21, 23–25, 29]. Wang [20] investigated directionality in the translation of metaphor, using eye-tracking, Translog, and retrospective reports, finding that directionality can significantly affect the relationship between processing types (ST processing, TT processing, and parallel processing), metaphor-related text types, and attention distribution patterns. Using eye-tracking and Translog, Whyatt et al. [21] studied the effect of directionality on the utilization of online resources. They found that online resources add cognitive effort when translators translating from L2 to L1. 23 Heeb [23] conducted a study involving a group of bidirectional translators and a group of unidirectional translators' views of direct and inverse translation, using introspective verbal protocols, and found no substantial differences in the self-concepts of the two groups. Pavlović and Jensen [29] compared professional translators' L1 and L2 translation processes, using eye-tracking. They found that some measures confirmed that L2 translation is cognitively more demanding than L1 translation, while others did not. Alves et al. [24] reported a study involving bidirectional translators and found that the cognitive effort involved in a translation process was not significantly different in regard to directionality.

Regarding the studies that focused on the use of online resources in the translation process, the theoretical underpinnings come from information studies, which state that information is needed when cognitive uncertainty arises [21]. It is assumed that when translator trainees translate into L1, they rely more on online resources. Whyatt et al. [21] found that online resources add cognitive effort when translators translate form L2 to L1. Pavlović [30] analyzed collaborative think-aloud protocols and noted that students relied more on online resources in L2-to-L1 translation and used the solutions they found more often than they did for L1-to-L2 translation. Muñoz et al. [31] reviewed neurocognitive studies and concluded that L2- to-L1 translation needs greater linguistic and extralinguistic processing.

## Perceived translation behaviors

Perceived translation behaviors refer to translators' own reporting of the strategies they employ during ST reading, TT drafting, and revising. Prior process-oriented studies have utilized Think Aloud Protocols (TAPs) and questionnaires to investigate cognitive translation processes from participants' perspectives [9, 32]. By means of oral translation tasks and TAPs, Lörscher [32] examined the translation strategies used by student translators for problem-solving. He identified several stages: recognizing, verbalizing, searching for solutions, and ultimately finding solutions to translation problems. Similarly, Krings [9] studied translation strategies of foreign language students using TAPs, translation tasks, and questionnaires. TAPs were used to identify translation problems and strategies. He [9] categorized student strategies into comprehension (like inference and reference use), equivalent retrieval (especially interlingual and intralingual associations), equivalent monitoring (e.g., comparing ST and TT), decision-making, and reduction.

**Cultural references in translation.** The translation theorists [1, 33, 34] are among those who concur that culture matters in translation. They emphasize that it is important to include cultural competence in translation training. It can be challenging to distinguish between culture and language because of their interconnectedness. Cultures are made up of norms, habits, experiences and regulations, and language is where all of these differences between cultures are most obvious [35].

According to Mailhac [36], culture is defined by the distance between the SL and the TL He defined a cultural reference as any SL reference distinct from TL culture, causing opacity for

the TL reader and causing comprehension difficulty. According to the definitions mentioned above, the implicit meaning of cultural aspects presents an interpretative challenge for a translator. Further, the translator perceives cultural components differently, as there are no specific criteria for determining what is and is not the best technique. As a result, the emphasis is on clear and accurate translation. This study focuses on cultural references–words or phrases from the source culture that lack equivalents or differ in the target culture. They are defined as "fixed, metaphorical phrases with non-negotiable meanings and forms" [37, p. 2]. Baker [2, p. 63] describes them as "frozen language patterns with little variation, often carrying meanings not deducible from their parts". These references have distinct denotations (literal meanings) or connotations (emotional responses) that may lack equivalents in the target culture. Scholars have used terms such as 'cultural terms' [1], 'culture-specific idioms' [2], 'cultural references' [4], and recently, 'cultural elements' [38].

Authors [39], Author [35] and Olk [4] investigated the cultural issue in terms of translators' behaviors in translation production. These studies focused on a translation product, and they ascribed the obstacles that the student translators experienced to the behaviors of the latter during the translation process. The authors found that students' poor product was due to comprehension problems, lack of analytical and production skills, inappropriate use of dictionaries, and lack of training in the use of translation strategies and techniques. Author [40] investigated the students' segmentation of cultural references when they translated them into Arabic, finding that the students' translations were inadequate and lacked fluency.

**Research questions.** The primary aim of the current study is to obtain empirical answers to the following questions:

1. Are there differences in actual translation behaviors (translation time, orientation time, revision time, online revision, online assistance, and scores) between direct translation and inverse translation of cultural references? If so, are these differences statistically significant?

2. Do the trainees' responses to questionnaire indicate differences in perceived translation behaviors between the direct translation and the inverse translation of cultural references?

## Method

This study used a computational model of human translation to investigate bidirectionality in English-Arabic-English translations of cultural references. The authors employed Translog-II software and a post-questionnaire to record trainees' actual and perceived translation behaviors during the translation of cultural references from L1 to L2 and L2 to L1. In order to measure and compare the trainees' actual and perceived translation behaviors, the study was conducted with a single group under two conditions (direct translation and inverse translation).

### Data collection

The authors utilized Translog software, translation tasks, and a post-questionnaire to capture participants' actual and perceived translation behaviors. The Translog software provided real-time data, while translation tasks demonstrated the participants' level of skills. The post-questionnaire gathered self-reported perceptions. This approach comprehensively examined participants' translation processes from both objective and subjective viewpoints.

### Translation task

The computer-based translation task consisted of two culture-based texts (one in English and the other in Arabic). Each text had 10 cultural references, which reflect Arabic and English.

The texts contained as many different cultural references as possible, which was the main criterion for selecting the text to be translated by the translator trainees. The second criterion was the authenticity of the text, which was intended to represent some features of the English culture.

The purpose of choosing the two texts is not to exhaustively cover cultural references, but to show translation trainees' behaviors when translating specific cultural references from English to Arabic and from Arabic to English. The text was kept short so as not to discourage the translators. The English text was taken from [41] *an*d [42], which target undergraduate translation training programs.

Both texts are descriptive articles. The English text is influenced by lexes of media while the Arabic text is influenced by religious lexes. These two texts were selected because lexes are strongly embedded in culture. The English and Arabic cultural expressions of the two texts are embedded in the English and Arabic cultures whose surface meanings are different from their intended meaning. Note that Arabic culture is strongly based on religion, while English culture does not prioritise some sources over others: they can relate to films, stories, political speeches, the Bible, political events, songs, etc. [43]. Furthermore, these two texts were used for didactic purposes and were found in two well-known translation textbooks.

## Questionnaire

The questionnaire was used to supplement the keylogging data, and was used to elicit trainees' views on the strategies they utilize during translation. The questionnaire items related to the following translation strategies: identifying problems; searching memory; contextualizing; guessing, using search engines, monolingual dictionaries, and bilingual dictionaries, and revising. Translator trainees were asked to tick the ones they used when translating a text. These strategies were based on [4] and [32] (S1 Appendix). The translator trainees were asked to tick the translation strategy they would use to translate the 20 English and Arabic cultural references. Hence, the selection of translation strategy in the questionnaire was confined to the cultural references targeted by the study and not the entire texts.

## Operationalization of variables

The study measured actual and perceived translation behaviors, cultural references, and translation quality. There were seven indicators for actual translation behaviors: translation time, orientation time, revision time, speed of translation, revision, and navigation behaviors. Translation quality was measured by applying [36] guidelines for acceptable translation. He proposed four criteria for acceptable translation of cultural references: the purpose of the text, the function of the cultural reference, the context, and the comprehension of the TL readership, which are summarized as two indicators: clarity and accuracy. Each test was scored out of 10. The authors assigned 1 mark if the translation of cultural references was considered adequate. In this study, a cultural reference is any expression that denotes any material, ecological, social, religious, political, linguistic, or emotional manifestation that can be attributed to a community [5]. The perceived translation behaviors(i.e., they are reported by the participants according to their self-perception) were operationalized by the translation strategies applied by translators when reading for comprehension, translating, and revising (identifying problems, searching memory, contextualizing, guessing, rereading, editing, and using bilingual dictionaries, monolingual dictionaries, and search engines) [4, 32].

The actual translation behaviors were captured by Translog software. The linear view of the program shows the orientation time (time spent on ST comprehension) via the icon of (Start) and revision time via the icon of (Stop), and the pauses between 'Start' and 'Stop' shows the drafting time. See Fig 1 below.

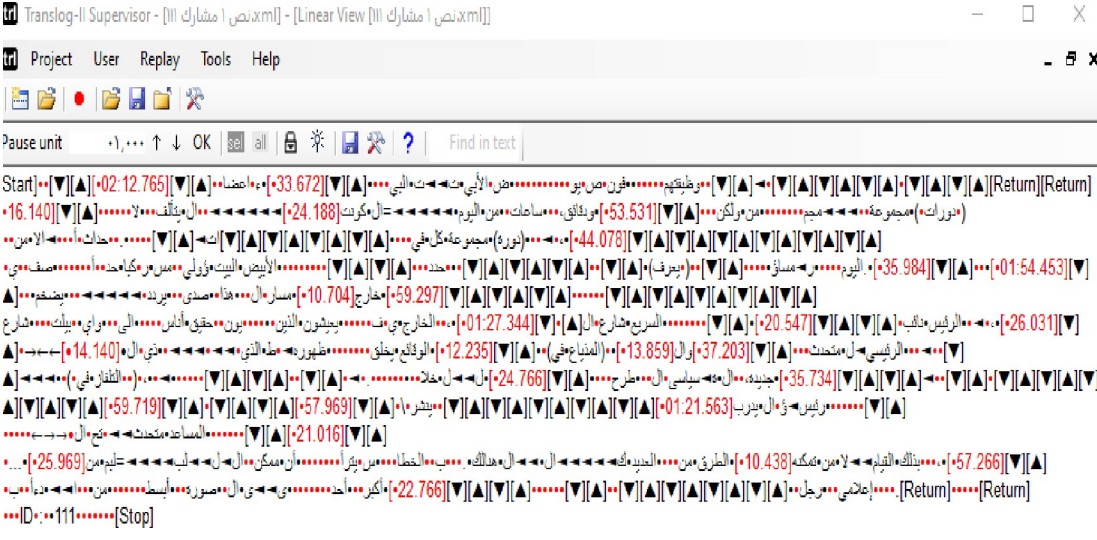

**Fig 1. Translog linear view.**

Revision behavior is recorded by Translog via the statistics function of the software, namely text elimination. The software counts the navigation behaviors via miscellaneous events, translation speed via the keystrokes per minute icon, and translation time via the duration icon. See Fig 2 below.

## Participants

The participants (N = 16) were Arabic females (ages: M = 19.57, SD = 0.06) who were enrolled in an Arabic-English translation program at one of Saudi Arabia's leading public universities. No males participated because of the gender segregation in the Saudi education system. The students are in the last year of the translation training program, having taken extensive courses in English-language skills in the first two years and translation skills and knowledge in the last two years. The sample size was small because the task involved Translog. Students required tutorial sessions in order to familarize themselves with the software and its functionalities.

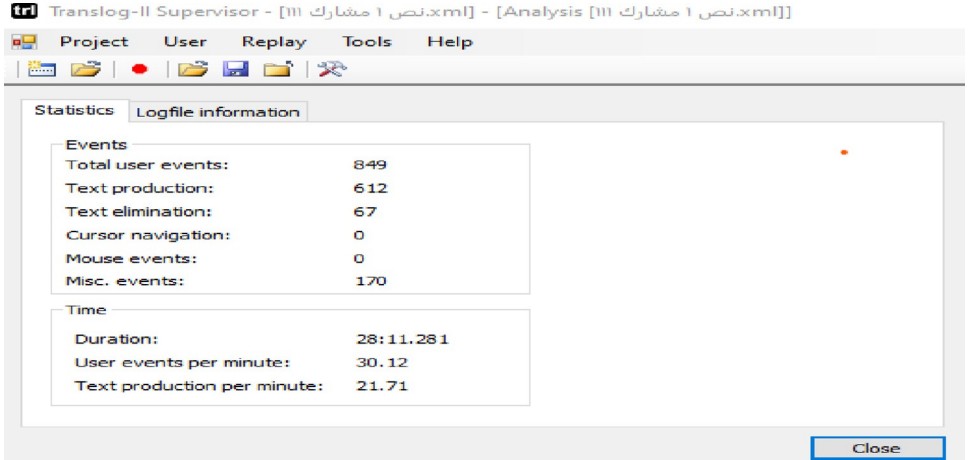

**Fig 2. The translog statistics.**

Moreover, certain students were unable to participate because of time constraints. Furthermore, the process of retrieving data from Translog is time-consuming, as the researcherz had to manually retrieve data for each individual participant from software.

To control for variability in typing speed that may affect the main variables of the study, participants' typing speed was assessed, and it was found to be comparable among the participants (M = 19.57, SD = 0.06). Participants received course credits for participating in the test. The experiment was conducted in the English language lab and confirmed that the students had no problems with their typing skills. Therefore, this variable did not affect student performance during translation. Potential participants had to meet several prerequisites to ensure comparability in L2 proficiency and educational background:

1. They are in the last year of the undergraduate translation program

2. The Participants completed an English proficiency measure: Brown's Cloze test, (M = 60, SD = 9.20, N = 18).

3. They have already taken part in the keylogging experiment and were familiar with the procedures.

4. Each participant provided informed consent to participate in this study.

5. Their typing speed is assessed (M = 19.57, SD = 0.06 per minute) so that the translation process is not affected by the typing variable,

## Informant consent

The participants gave verbal consent to participate in the study after being assured that their participation was voluntary and that they had the right to withdraw their consent at any time. Also, their anonymity was assured and only de-identified data would be used in publications, ensuring that their personal information and identity remain confidential. Moreover, participants were told that the data collected during the study would be used solely for research purposes and could be published in scientific journals or presented at academic conferences. The data would be securely stored and accessible only to authorized researchers. The recruitment period was from 11/07/2022to 5/10/2022.

## Data analysis

The data were analyzed quantitatively using the Statistical Package for the Social Sciences (SPSS 27). The researchers used the Wilcoxon test, a counterpart to the paired t-test. The Wilcoxon test measures the differences in actual translation behaviors and perceived translation behaviors for the direct translation and inverse translation of cultural references, and is appropriate when the sample is small and the distribution of the data is skewed [44]. Due to the skewness of data, the researchers determined the median to show the central tendency of the data since it is better than the mean for data that exhibit skewness [44].

## Translation brief

The participants were asked to render the texts in such a way that TL readers could understand the SL culture; furthermore, their translations were intended for the general Arabic readership. Newmark [1] identified three types of TL readers (expert, educated generalist, and uninformed) that translators must address in their translation. Furthermore, the translator trainees were given the background of the texts they would be translating

## Assessment criteria

Mailhac's [36] standards for accepted translation were adopted for the study. The standards include the purpose of the text, the role of the cultural reference, context, and TL readership understanding. Based on [36] standards, the researchers developed a rubric for two indicators: accuracy and clarity. This rubric was examined by two raters to determine its suitability for the study. Two translation experts assessed the students' translations according to their accuracy and clarity; a score of 0.5 was allocated to each indicator.

| Indicators of acceptability | | | | | | |
| --- | --- | --- | --- | --- | --- | --- |
| Accuracy | Definition | Score | Clarity | Definition | Score |
| | conveying the original message | 0.5 | | Clarity for TL readers | 0.5 |

# Results

This section addresses the research questions regarding the differences between direct and inverse translation in terms of actual and perceived translation behaviors. Below, empirical responses to these questions are provided.

## Actual translation behaviors

**Temporal effort.** The analysis showed differences between direct and inverse translation in terms of duration, reading time, and revision time. Orientation time (Mdn = 138 seconds) and total translation time (Mdn = 46.2050 minutes) were greater for direct translation than for orientation (Mdn = 42.50 seconds) and total translation time (Mdn = 45.7800 minutes) in inverse translation. Similarly, direct translation had longer revision time (Mdn = 32.00 seconds) than inverse translation (Mdn = 20.06 seconds). See Table 1.

The analysis of ranks obtained by a Wilcoxon signed-ranked test revealed that positive ranks for direct translation were higher than those for inverse translation in terms of entire translation time. Conversely, in the cases of revision time and reading time, the number of negative ranks was greater for inverse translation than for direct translation. See Table 2.

The Wilcoxon-signed ranks test showed differences between direct translation and inverse translation of cultural references in terms of time spent on orientation, drafting, and revision, but they are not statistically significant. The statistically significant differences were detected in orientation time between direct and inverse translation (n = 16, z = -2.482, p < .05) with a moderate effect size (r = .620). Note that the effect size is counted in the Wilcoxon signed-Rank Test by dividing the value of 'z' by the square toot of the number of participants $r = \frac{z}{\sqrt{N}}$. (Mellinger and Hanson, 2017).

See Table 3.

## Online revision

Keystroke data revealed the translator trainees' online revisions keystrokes (trainees translate and revise the text at the same time) for direct translation and inverse translation. Analysis of

**Table 1. Median and mean of product and process behaviors.**

| Variable | Direct translation | Inverse translation |
| --- | --- | --- |
| | Median | Median |
| Reading time (orientation) | 138 | 42.50 |
| Post-editing (Revision) | 32 | 20.06 |
| Duration | 46.2050 | 45.7800 |

**Table 2. Negative and positive ranks.**

| | | N | Mean Rank | Sum of Ranks |
|---|---|---|---|---|
| Whole time of IT–DT | Negative Ranks | 7 | 8.43 | 59.00 |
| | Positive Ranks | 9 | 8.56 | 77.00 |
| | Ties | 0 | | |
| | Total | 16 | | |
| Revision time in IT–DT | Negative Ranks | 9 | 7.83 | 70.50 |
| | Positive Ranks | 6 | 8.25 | 49.50 |
| | Ties | 1 | | |
| | Total | 16 | | |
| Orientation time in IT-DT | Negative Ranks | 12 | 9.67 | 116.00 |
| | Positive Ranks | 4 | 5.00 | 20.00 |
| | Ties | 0 | | |
| | Total | 12 | | |

**Table 3. Wilcoxon-signed ranks test.**

| Measures | Z | P-Value | |
|---|---|---|---|
| Orientation time | -2.482 | .013 | .620 |
| Duration | -.465-[b] - | .642 | .114 |
| Post-editing time | -.596-[c] | .551 | .149 |

**Table 4. Median of online revision keystrokes.**

| Variable | Direct translation | Inverse translation |
|---|---|---|
| Online revision keystrokes | Median | Median |
| | 153.00 | 279.50 |

**Table 5. Ranks of online revision keystrokes.**

| | | N | Mean ranks | Sum of ranks |
|---|---|---|---|---|
| Online revision IT–DT | Negative Ranks | 3 | 2.00 | 6.00 |
| | Positive Ranks | 13 | 10.00 | 130.00 |
| | Ties | 0 | | |
| | Total | 16 | | |

data revealed translator trainees had different online revision behaviors when engaged in direct translation (Mdn = 153.0 keystrokes) and inverse translation (Mdn = 279.50 keystrokes), indicating that the number of online revision keystrokes for inverse translation is greater than for direct translation. See Table 4.

The Wilcoxon signed-ranked test showed more online revision keystrokes in inverse translation than in direct translation. Of the 16 translator trainees, 13 used more online revision keystrokes in inverse translation. See Table 5.

The Wilcoxon matched-pairs signed rank indicated statistically significant differences in online revision between direct translation and inverse translation in favor of inverse translation (n = 16, z = -3.206 < .05 with a large effect size, r = .889) See Table 6.

## Production speed

Analysis of data revealed that participants translated at different speeds when translating cultural references in direct translation and inverse translation. The results shown in Table 7 indicate that translation speed was greater for inverse translation (Mdn = 34.5 keystrokes a minute) than direct translation (Mdn = 21.50 keystrokes per minute).

The Wilcoxon signed-ranked test revealed that the ranks of inverse translation (positive ranks) were higher than the ranks of direct translation (negative ranks). Of the 16 participants, 13 exhibited a higher production speed in inverse translation compared to direct translation. See Table 8.

The Wilcoxon matched-pairs signed rank indicated statistically significant differences in the speed of production between direct translation and inverse translation in favor inverse translation (n = 16, z = —3.068< .05 with a large effect size, r = .767) See Table 9.

## Navigation behaviors

Translog revealed that the participants relied heavily on online resources for both inverse translation (Mdn = 333.00) and direct translation (Mdn = 212), but they relied more heavily on online resources for direct translation than inverse translation. See Table 10.

Table 6. Wilcoxon-signed ranks test.

| Measures | Z | P-Value | R |
|---|---|---|---|
| Orientation time | -3.206 | .001 | .889 |

Table 7. Median of speed of production.

| Variable | Direct translation | Inverse translation |
|---|---|---|
| Speed of production | Median | Median |
|  | 21.50 | 34.5 |

Table 8. Ranks of speed of production.

|  |  | N | Mean Rank | Sum of Ranks |
|---|---|---|---|---|
| Speed of IT–speed of DT | Negative Ranks | 2 | 3.00 | 6.00 |
|  | Positive Ranks | 13 | 8.77 | 114.00 |
|  | Ties | 1 |  |  |
|  | Total | 16 |  |  |

Table 9. Wilcoxon-signed ranks test of speed of production.

| Measures | Z | P-Value | R |
|---|---|---|---|
| Orientation time | -3.068 | .002 | .767 |

Table 10. Median of navigation behaviors.

| Variable | Direct translation | Inverse translation |
|---|---|---|
| Navigation of behaviors | Median | Median |
|  | 333.00 | 212 |

The Wilcoxon signed-ranked test indicated higher negative ranks for direct translation compared to positive ranks for inverse translation. The results showed that 13 participants relied more heavily on online resources for direct translation than for inverse translation. See Table 11.

The Wilcoxon matched-pairs signed rank indicated that the differences in navigation behaviors between direct translation and inverse translation are not statistically significant = 16, $z = -672 < . .501$ with a large effect size, $r = .186$. See Table 12.

## Translation scores

The Translog report revealed that the translator trainees scored higher on inverse translation (Mdn = 7.250) than on direct translation (Mdn = 6.000). See Table 13.

The Wilcoxon signed-ranked test showed more positive ranks for inverse translation than negative ranks for direct translation. Table 14 shows that 15 out of 16 participants scored higher on inverse translation.

Results showed statistically significant differences in the trainees' scores between direct and inverse translation in favor of inverse translation ($n = 16$, $z = -3.346 - < .05$ with a large effect size, $r = . 836$). See Table 15.

**Table 11. Ranks of navigation behaviors.**

|  |  | N | Mean Rank | Sum of Ranks |
|---|---|---|---|---|
| Navigation in IT–navigation in DT | Negative Ranks | 10 | 8.10 | 81.00 |
|  | Positive Ranks | 6 | 9.17 | 55.00 |
|  | Ties | 0 |  |  |
|  | Total | 16 |  |  |

**Table 12. Wilcoxon-signed ranks test.**

| Measure | Z | P-Value | R |
|---|---|---|---|
| navigation behaviors | -.672 | .501 | .186 |

**Table 13. Median translation scores.**

| Variable | Direct translation | Inverse translation |
|---|---|---|
| Scores | Median | Median |
|  | 6.000 | 7.250 |

**Table 14. Ranks of scores in direct and inverse translation.**

|  |  | N | Mean Rank | Sum of Ranks |
|---|---|---|---|---|
|  | Total | 16 |  |  |
| Scores in IT–scores in DT | Negative Ranks | 1 | 4.00 | 4.00 |
|  | Positive Ranks | 15 | 8.80 | 132.00 |
|  | Ties | 0 |  |  |
|  | Total | 16 |  |  |

**Table 15. Wilcoxon-signed ranks test.**

| Measures | Z | P-Value | R |
|---|---|---|---|
| Orientation time | -3.346- | .001 | . 836 |

**Table 16. Translation strategies utilized when translating cultural references in direct translation.**

| Translation perceived behaviors | Direct translation | | Inverse translation | |
|---|---|---|---|---|
| | N | Percent of cases | N | Percent of cases |
| Identifying problems | 36 | 6.2% | 15 | 5.3% |
| Searching memory | 28 | 4.8% | 43 | 15.2% |
| Guessing | 81 | %14.1 | 49 | 17.3% |
| Contextualizing | 78 | 13.5% | 37 | 13.1% |
| Rereading | 59 | 10.2% | 9 | 3.1% |
| Using search engines | 117 | 20.3% | 42 | 14.8% |
| Using a bilingual dictionary | 110 | 19.1% | 30 | 10.6% |
| Using a monolingual dictionary | 38 | 6.6% | 13 | 4.6% |
| Revising | 27 | 4.7% | 44 | 15.6% |
| Total frequencies | 574 | 100% | 282 | 100% |

## Translator trainees' perceived translation behaviors in direct and inverse translation

Trainees' questionnaire responses showed differences between direct and inverse translation in terms of perceived translation behaviors. For direct translation, students used bilingual dictionaries (19.1%) and search engines (20.3%) more than for inverse translation. However, for inverse translation, there was greater reliance on guessing and memory retrieval than for direct translation. It was found that the least frequently-used behaviors when translating the cultural references in direct translation were revision (4.7%), searching memory (4.8%), guessing (14.1), contextualizing (13.5%), and using a monolingual dictionary (6.65%); however, for inverse translation, monolingual dictionary, identifying problems and re-reading were the least-used strategies. See Table 16.

## Discussion

The students' translations of English text into Arabic (direct translation) and Arabic text into English (inverse translation) revealed different translation behaviors in terms of temporal effort (translation time, orientation time, and end revision time), online revision, use of online resources, and speed of translation. The students' responses to the questionnaire revealed their use of internal strategies and external strategies when translating from English to Arabic (direct translation) and Arabic into English (inverse translation). The main findings of this study are presented and discussed below.

### Temporal effort

Findings revealed both direct translation and inverse translation are time-consuming. There were differences in translation time between inverse translation and direct translation, but these differences are not statistically significant despite the greater amount of translation time required when translating the Arabic text into English.

In regard to the time spent on orientation (ST reading), statistically significant differences were detected between direct translation and inverse translation, with students spending more

time on orientation during the direct translation task. This finding indicated that the trainees spent more time on reading the English text than they did on the Arabic text. This could be attributed to the fact that translator trainees find reading in their own native language easier than reading a text in a second language. This observation aligns with [45], who found that L2 translation requires greater linguistic and extralinguistic processing than L1.

Concerning the time spent on final revision, there were differences in revision time between direct and inverse translation, with revision of direct translation taking more time, although these differences were not statistically significant. Translator trainees took more time to revise the English text than the Arabic text. The results highlighted the variations in reading and revision processes. Reading the Arabic text was quicker than the English text, whereas revising the English text took longer than did the Arabic text. This finding implies that translator trainees require more practice in reading and revision in the English language. Similarly, Pavlović and Jensen [29] found that that processing the L2 text requires more cognitive effort than L1. 20 Wang [21] found that directionality can have a significant effect on the relationship between ST processing, TT processing, and parallel processing. Teachers delivering translation courses should reflect on these findings and train students to manage the translation process effectively by providing appropriate material and activities in class. They should design the course content so that students can develop skills to tackle the three phases of translation (i.e., reading, drafting, and revision) in both directions: inverse and direct translation. According to Authors [7], the translation class should develop students' skills in reading, writing, and revision by integrating these skills into a translation task [7]. Teachers should not only target the final product of translation but also evaluate students' reading comprehension, writing, and revision skills. Students should be trained on how to coordinate reading, drafting, and revision to produce a satisfactory translation. Specialized courses (e.g., revision, reading, and drafting in translation) should be offered in translation programs [46, 47].

## Online revision

Findings revealed statistically significant differences in online revision between direct translation and inverse translation, indicating that the translator trainees relied more heavily on online revision during inverse translation (Arabic to English). In other words, translator trainees' online revision of TT (translating from Arabic into English) during drafting occurs mainly during inverse translation, which may hamper TT production. Similarly, Dragsted and Carl [48] discovered that novice translators overused online revision at the expense of end revision. Trainees should be trained to manage each stage of the translation process beginning with ST comprehension, followed by TT production and, ultimately, TT revision. During this process, trainees need to effectively coordinate ST reading and TT production (drafting) to achieve translation quality. It is essential that revision take place after drafting has been completed. Both should not be attempted simultaneously as ongoing revision could impose a significant cognitive burden on the translator. Hence, translation programs should emphasize the importance of revision being done at the end of the drafting process. Several scholars [46, 47] have proposed revision competence models. Therefore, translation classes and textbooks should prioritize revision in their teaching methods. Mossop [49] should be used by translation teachers when designing translation activities. Trainees must be trained to become translators and revisers by incorporating revision tasks into their translation tasks.

## Use of online resources

Our findings revealed non-statistically significant differences in navigation behaviors among translator trainees when translating cultural references from English to Arabic (direct

translation) and Arabic into English (inverse translation). The Translog report showed that the navigation behaviors are greater during direct translation than inverse translation, but these differences are not statistically significant. The students' questionnaire responses revealed that trainees depended more on online resources (bilingual dictionaries and search engines) during direct translation than inverse translation, which supplement Translog data. Similarly, Whyatt et al. [21] found that online resources add cognitive effort when translators work into their L2 (L2-L1). Pavlović [30] analyzed collaborative think-aloud protocols and noted that students relied more on online resources for L2 translation, and used the solutions they found more often than for L1 translation. Muñoz et al. [31] reviewed neurocognitive studies and concluded that L2 translation needs greater linguistic and extralinguistic processing.

The students' frequent navigation keystrokes in both direct translation and inverse translation revealed their heavy reliance on external strategies (i.e., search engines and online dictionaries) to solve comprehension and production problems. Huang et al. [50] established the association between navigation keystrokes and search engines and online dictionaries. Similarly, Hvelplund [51] and Enríquez Raído [52] found that the students' relied heavily on digital sources. On the other hand, professional translators relied heavily on internal knowledge (i.e., context) to solve translation problems, whereas novice translators relied heavily on external tools (e.g., online dictionaries). The ideal situation for using digital sources is when they are used in context [39, 53]. Students should be trained to use monolingual dictionaries, bilingual dictionaries, and search engines (mining the Web) appropriately in accordance with the requirements of a text. In some cases, students must go beyond dictionaries and use search engines to find the definition of a term or translations available on the Web.

## Speed of production

Results showed statistically significant differences between direct translation and inverse translation in terms of participants' speed. The trainees were slower in direct translation than in inverse translation, which demonstrated that the trainees had more difficulty with ST comprehension and production in direct translation than inverse translation tasks. This finding partially aligns with [29] who found that some measures confirmed that L2 translation is cognitively more demanding than L1 translation, while others did not. What can be concluded here is that some measures confirmed that inverse translation is more effortful such as online revision, orientation time while other measures (translation scores and translation production) revealed that direct translation requires more effort. Such findings indicate that the nature of the task and the translator's level of competence in the first and the second languages determine the temporal, cognitive and technical efforts.

## Translation quality

Findings revealed statistically significant differences in scores between direct translation and inverse translation, which showed that the trainee's translation performance in inverse translation is better than in direct translation, which might be attributed to trainees' understanding of Arabic texts, which simplified their rendition, in addition to the use of internal transition strategies (contextualization, and searching memory). This finding may be different from those studies that found inverse translation to be more effortful. However, this finding aligned partially with [29] who compared professional translators' L1 and L2 translation processes, using eye-tracking, finding that some measures confirmed that L2 translation is cognitively more demanding than L1 translation. This finding may show that the trainees found useful resources for inverse translation, which simplified their production. However, this does not mean that inverse translation did not show deviant translation and translation loss; rather,

inverse translations are better than direct translations. In both texts, trainees' inadequate renditions revealed that they did not give due attention to reading for comprehension and revising the end product, suggesting a lack of skills in using digital sources and internal translation strategies to achieve an acceptable rendition. Similarly, Lörscher [32], Carl et al. [54] and Sharmin et al. [55] argued that students' translation behaviors during the entire translation process will have either a positive or negative effect on translation quality.

Translation training programs should include specific assessment criteria for both direct and inverse translation to ensure the objectivity and reliability of translation assessment, drawing on the latest models of translation quality assessment. Specific assessment criteria will guide teachers and help to ensure that they assess more fairly and accurately, and will be a means of providing feedback to students who can refer to the criteria when they are engaged in translations.

Despite trainees' temporal (time) and technical efforts (revision and navigation keystrokes), they were unable to properly render many cultural references into Arabic, as evidenced by the mean scores and trainees' translations. These findings corroborated Translog data (short time spent on reading and short post-writing revision) and the questionnaire responses, which revealed that the students for the most part did not use internal translation strategies such as problem identification, re-reading, and contextualization. According to Gopferich [56], translators' inability to manage the translation process prevents them from producing a high-quality product. On the other hand, Author [35] discovered that relying solely on internal translation strategies leads to comprehension and production issues.

### Perceived translation behaviors direct and inverse translation

The findings showed that during direct translation, trainees relied more on online resources than they did for inverse translation. Conversely, for inverse translation, the trainees tended to use internal strategies such as contextualization and memory retrieval more so than for direct translation. These outcomes align with the Translog data, confirming that students utilized online resources far more for direct translation than for inverse translation. See S1 File.

Similar to the keylogging data, the data for the perceived translation behaviors revealed the trainees' reliance on online resources for direct translation. Using digital sources without first identifying problematic areas in a text may result in a departure from the SL meaning. The use of online resources should be in the light of context [35]. In this regard, translation programs should teach students how to solve comprehension and production problems using digital sources and other comprehension techniques. Students should be taught the what, when, how, and why to employ a specific translation strategy to solve a comprehension or production problem. Trainees' use of multiple translation strategies (e.g., searching memory, contextualizing, and using monolingual and bilingual dictionaries) when translating the one cultural reference indicated that they were unsure about the best strategy to use to solve the translation problem. This finding showed the students' lack of training in the use of digital sources and internal translation strategies. Translation teachers should ensure that students learn how to achieve a balance between the use of digital sources and internal translation strategies (i.e., searching memory, rereading, contextualizing, guessing, and revising) to overcome comprehension and production problems.

### Conclusion

This study investigated the various stages of the translation process and measured the behavior of translator trainees when engaged in direct and inverse translation to show the areas of translation training that require more focus and improvement. The analysis of students' translation

behaviors when translating cultural references revealed their weaknesses in the management of translation stages, and their lack of cultural competence in both direct and inverse translation. The training programs should be designed to improve the trainees' translation competence in different stages of translation process in both direct and inverse translation.

The robustness of this study is primarily derived from an investigation of the various stages of the translation process in bidirectional translation of cultural references through the use of Translog software. The software not only serves as a valuable resource for translation educators but also provides an innovative approach to assess students' performance and activities across different translation phases. The study follows a process-oriented translation approach, enabling an in-depth exploration of the intricate facets of the translation process, which facilitates the identification of specific areas where trainees may encounter difficulties, thereby contributing to understanding their learning needs. Nonetheless, certain weaknesses are worth noting. The study exhibits a limited emphasis on the students' translations. Moreover, the small sample size employed in the research poses a constraint on the generalizability of the findings. Expanding the participant pool and incorporating a more diverse range of texts could enhance the study's robustness and validity. These considerations should be addressed in future research endeavors to further enrich our understanding of this critical subject matter.

## Supporting information

**S1 Appendix.**
(DOCX)

**S1 File.**
(XLSX)

## Author Contributions

**Conceptualization:** Mutahar Qassem.

**Data curation:** Buthainah M. Al Thowaini.

**Formal analysis:** Mutahar Qassem.

**Funding acquisition:** Mutahar Qassem.

**Investigation:** Buthainah M. Al Thowaini.

**Methodology:** Mutahar Qassem.

**Project administration:** Buthainah M. Al Thowaini.

**Resources:** Mutahar Qassem.

**Software:** Mutahar Qassem.

**Supervision:** Buthainah M. Al Thowaini.

**Validation:** Mutahar Qassem.

**Visualization:** Buthainah M. Al Thowaini.

**Writing – original draft:** Mutahar Qassem.

**Writing – review & editing:** Buthainah M. Al Thowaini.

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
