## [Decision Letter · Decision Letter 0]

20 Aug 2023

PONE-D-23-24645Bi-directionality in translating culture: Understanding translator trainees’ actual and perceived behaviorsPLOS ONE

Dear Dr. Qassem,

Thank you for submitting your manuscript to PLOS ONE. After careful consideration, we feel that it has merit but does not fully meet PLOS ONE’s publication criteria as it currently stands. Therefore, we invite you to submit a revised version of the manuscript that addresses the points raised during the review process.

We look forward to receiving your revised manuscript.

Kind regards,

Anastassia Zabrodskaja, Ph.D.

Academic Editor

PLOS ONE

Journal Requirements:

   "The project is funded by the Deanship of Scientific Research at Najran University"

4. Please ensure that you include a title page within your main document. You should list all authors and all affiliations as per our author instructions and clearly indicate the corresponding author.

6. Please include a separate caption for each figure in your manuscript.

7. We note that Figures 1 and 2 in your submission contain copyrighted images. All PLOS content is published under the Creative Commons Attribution License (CC BY 4.0), which means that the manuscript, images, and Supporting Information files will be freely available online, and any third party is permitted to access, download, copy, distribute, and use these materials in any way, even commercially, with proper attribution. For more information, see our copyright guidelines: http://journals.plos.org/plosone/s/licenses-and-copyright.

a. You may seek permission from the original copyright holder of Figures 1 and 2 to publish the content specifically under the CC BY 4.0 license. 

8. We note you have included a table to which you do not refer in the text of your manuscript. Please ensure that you refer to Table 4 and 5 in your text; if accepted, production will need this reference to link the reader to the Table.

Additional Editor Comments:

Please read all reviewers' comments and revise your paper carefully. 

Reviewers' comments:

Reviewer's Responses to Questions

**Comments to the Author**

1. Is the manuscript technically sound, and do the data support the conclusions?

Reviewer #1: Yes

Reviewer #2: Partly

Reviewer #3: Yes

2. Has the statistical analysis been performed appropriately and rigorously? 

Reviewer #1: Yes

Reviewer #2: No

Reviewer #3: No

3. Have the authors made all data underlying the findings in their manuscript fully available?

Reviewer #1: No

Reviewer #2: Yes

Reviewer #3: Yes

4. Is the manuscript presented in an intelligible fashion and written in standard English?

Reviewer #1: Yes

Reviewer #2: Yes

Reviewer #3: Yes

5. Review Comments to the Author

Reviewer #1: In the process of reviewing the submitted manuscript, it is evident that the authors have invested significant time and effort in employing a variety of data collection and analysis tools. While the article demonstrates a commendable level of rigor, some areas require attention. Specifically, several in-text citations warrant careful verification to ensure alignment with the reference list. Additionally, one table lacks proper numbering, necessitating correction for coherence. Language-related issues have been identified in the conclusion and Text 2, affecting the overall clarity and fluency. All in all, the manuscript holds great promise and merits publication. Its comprehensive utilization of diverse data collection and analysis tools attests to the authors' dedication.

Reviewer #2: ID: PONE-D-23-24645

Title: Bi-directionality in translating culture: Understanding translator trainees’ actual and perceived behaviors

Thank you for providing a chance to review this manuscript.

Detailed information:

Abstract

Overall: 1) What is the research background of this article? What was the specific study design? 2) Please add specific statistical analysis methods. 3) The results section should list all key values, not just a lack of evidence in language. 4) If the journal does not have special requirements, I recommend presenting the abstract section in order of background/objective, methods, results, conclusions.

1. Introduction

Paragraph 1, page 2: What are SL and TL?

1.2 Computational Model of Human Translation: I don't quite understand what you're trying to say in this paragraph. First of all, if it is to explain how Translog software records the behavior of students by tracking the behavior data of translators to illustrate the feasibility of this method, I think it is more appropriate to put it in the method part. Second, "But can translator trainees use these resources well? It depends on the search skills of the translator and the availability of information on the web...” What is the point of this paragraph?

1.3 Theoretical Framework: It is recommended to summarize and simplify the content.

3. Method: Note the serial number.

3.4 Participants: There were 16 subjects in the study? Is the sample representative enough? I don't think your sample size meets the requirements. If yes, please provide corresponding theoretical support. And please improve the inclusion and exclusion criteria of study subjects.

3.5. Informatnt consent: Please supplement the ethical review.

Overall: Please pay attention to the error in writing even the basic title number. And, the order of the entire method section is very chaotic, please follow the research design approach.

4. Results

Please unify all table formats into a three-line table, paying special attention to Table 5 and Table 6.

5. Discussion

1) Where is Appendix 3? 2) Please describe the strengths and limitations of this study, and how the article will help future research.

6. Conclusion

It is recommended to simplify the content.

Overall, 1) your study was limited to female translators in the Arab region, and the sample size was only 16. I don't think the sample meets the requirements. 2) The innovation of this study is not enough, and many details were not paid attention to, making the detailed description too brief. 3) Your charts and tables are not standardized enough, like a "draft". And the entire text is too verbose, please pay attention to extracting key information. 4) Refine and simplify your introduction, I really can't read everything in its entirety. 5) Note the text format and numbering of the article.

Thank you and my best,

Your reviewer

Reviewer #3: The study investigated the behaviors of translator trainees in different translation directions (L2 to L1 and L1 to L2) while translating cultural references between English and Arabic. The focus was on how the directionality of translation impacts their actions, including orientation and revision behaviors, translation quality, and use of online resources. The study was well designed overall. However, the analysis of the data with a small number of participants (n = 16) should be checked by statisticians. The small sample size may require the use of other statistical procedures, such as non-parametric statistics tests.

6. PLOS authors have the option to publish the peer review history of their article (what does this mean?). If published, this will include your full peer review and any attached files.

Reviewer #1: No

Reviewer #2: No

Reviewer #3: No

---

## [Author Response · Author response to Decision Letter 0]

27 Sep 2023

I hope this email finds you well. I am writing to submit the revised version of our manuscript titled " Bi-directionality in translating culture: Understanding translator trainees’ actual and perceived behaviors" following the invaluable feedback from the reviewers. We sincerely appreciate the reviewers' time and effort in providing constructive comments, which have significantly contributed to improving the quality of our work. Please find below our responses to each of the reviewers' comments. See the responses in the attachment.

---

## [Decision Letter · Decision Letter 1]

16 Oct 2023

Bi-directionality in translating culture: Understanding translator trainees’ actual and perceived behaviors

PONE-D-23-24645R1

Dear Dr. Qassem,

We’re pleased to inform you that your manuscript has been judged scientifically suitable for publication and will be formally accepted for publication once it meets all outstanding technical requirements.

Kind regards,

Anastassia Zabrodskaja, Ph.D.

Academic Editor

PLOS ONE

Additional Editor Comments (optional):

Reviewers' comments:

Reviewer's Responses to Questions

**Comments to the Author**

1. If the authors have adequately addressed your comments raised in a previous round of review and you feel that this manuscript is now acceptable for publication, you may indicate that here to bypass the “Comments to the Author” section, enter your conflict of interest statement in the “Confidential to Editor” section, and submit your "Accept" recommendation.

Reviewer #1: All comments have been addressed

Reviewer #3: All comments have been addressed

2. Is the manuscript technically sound, and do the data support the conclusions?

Reviewer #1: Yes

Reviewer #3: Yes

3. Has the statistical analysis been performed appropriately and rigorously? 

Reviewer #1: Yes

Reviewer #3: Yes

4. Have the authors made all data underlying the findings in their manuscript fully available?

Reviewer #1: Yes

Reviewer #3: Yes

5. Is the manuscript presented in an intelligible fashion and written in standard English?

Reviewer #1: Yes

Reviewer #3: Yes

6. Review Comments to the Author

Reviewer #1: In this revised version of the paper, the authors have made it more accessible to readers by strengthening the organization and flow of the manuscript. Also, they have addressed the majority of the reviewer comments and suggestions, demonstrating a clear commitment to enhancing the quality of the work. Therefore, I recommend the acceptance of the revised manuscript for publication.

Reviewer #3: The authors have carefully considered my suggestions and made appropriate changes to the manuscript.

7. PLOS authors have the option to publish the peer review history of their article (what does this mean?). If published, this will include your full peer review and any attached files.

Reviewer #1: No

Reviewer #3: No

---

## [Editor Report · Acceptance letter]

19 Oct 2023

PONE-D-23-24645R1 

Bi-directionality in translating culture: Understanding translator trainees’ actual and perceived behaviors 

Dear Dr. Qassem:

I'm pleased to inform you that your manuscript has been deemed suitable for publication in PLOS ONE. Congratulations! Your manuscript is now with our production department. 

Kind regards, 

on behalf of

Professor Anastassia Zabrodskaja 

Academic Editor

PLOS ONE